# Cholesterol Redistribution in Pancreatic β-Cells: A Flexible Path to Regulate Insulin Secretion

**DOI:** 10.3390/biom13020224

**Published:** 2023-01-24

**Authors:** Alessandra Galli, Anoop Arunagiri, Nevia Dule, Michela Castagna, Paola Marciani, Carla Perego

**Affiliations:** 1Department of Pharmacological and Biomolecular Sciences (DiSFeB), Università degli Studi di Milano, 20134 Milan, Italy; 2Division of Metabolism, Endocrinology & Diabetes, Department of Internal Medicine, University of Michigan, Ann Arbor, MA 48106, USA

**Keywords:** insulin biosynthesis and secretion, cholesterol homeostasis, cholesterol trafficking, β-cell dysfunction, type 2 diabetes

## Abstract

Pancreatic β-cells, by secreting insulin, play a key role in the control of glucose homeostasis, and their dysfunction is the basis of diabetes development. The metabolic milieu created by high blood glucose and lipids is known to play a role in this process. In the last decades, cholesterol has attracted significant attention, not only because it critically controls β-cell function but also because it is the target of lipid-lowering therapies proposed for preventing the cardiovascular complications in diabetes. Despite the remarkable progress, understanding the molecular mechanisms responsible for cholesterol-mediated β-cell function remains an open and attractive area of investigation. Studies indicate that β-cells not only regulate the total cholesterol level but also its redistribution within organelles, a process mediated by vesicular and non-vesicular transport. The aim of this review is to summarize the most current view of how cholesterol homeostasis is maintained in pancreatic β-cells and to provide new insights on the mechanisms by which cholesterol is dynamically distributed among organelles to preserve their functionality. While cholesterol may affect virtually any activity of the β-cell, the intent of this review is to focus on early steps of insulin synthesis and secretion, an area still largely unexplored.

## 1. Introduction

Pancreatic β-cells are responsible for maintaining the blood glucose levels within a very narrow range by synthesizing and secreting the major hypoglycemic hormone, insulin. Despite reports indicating insulin expression in tissues, such as the brain or liver [1,2], the pancreatic β-cells remain the major source of insulin biosynthesis and secretion; therefore, their dysfunction and/or death result in decompensated glucose homeostasis and diabetes development [3,4]. Diabetes indicates a group of heterogeneous metabolic disorders characterized by fasting hyperglycemia; among them, type 2 diabetes (T2D) deserves particular attention given its exponential growth worldwide; it is expected to reach almost 700 million cases in 2045 [5]. T2D is often associated with a cluster of lipid abnormalities—which include low HDL (high-density lipoproteins) and increased levels of triglycerides and remnant lipoproteins (LDL, low-density lipoproteins, and VLDL, very low-density lipoproteins)—involved in cholesterol transport [6]. Diabetic dyslipidemia not only increases the risk of cardiovascular events but also contributes to the β-cells dysfunction [7,8]. Indeed, cholesterol can be either “friend or foe” of pancreatic β-cells as appropriate cholesterol levels are necessary for β-cell functions, insulin biogenesis and secretion; however, inadequate cholesterol levels strongly affect the ability of β-cells to synthetize and secrete the proper amount of insulin, while an excess can lead to β-cell demise [9,10,11,12,13,14].

Through this review, we aim to provide the most recent perspectives on how cholesterol homeostasis could influence insulin biosynthesis and secretion and, thus, the overall function of pancreatic β-cells.

## 2. Insulin Biosynthesis, Sorting and Secretion

The insulin gene, located on the chromosome 11, codes for a 110-amino acid precursor, called pre-proinsulin, which undergoes multiple enzymatic reactions to yield insulin. Insulin biogenesis and secretion are multi-step processes that are tightly controlled by nutrient availability. Although several nutrients may control insulin secretion, glucose is the well-characterized one [15,16]. To sustain β-cell insulin secretion, glucose not only promotes hormone release but also increases the hormone transcription and translation. Interestingly, insulin release begins in seconds after the increase of glucose levels, while mRNA translation and transcription are activated from minutes to hours [17]. The main steps of insulin production and secretion are summarized in Figure 1. 

### 2.1. Insulin Biogenesis, Processing and Storage into Secretory Granules

A network of transcription factors, including PDX-1 (pancreatic and duodenal homeobox), BETA2/NeuroD1 (BETA2/Neurogenic differentiation 1) and MafA (v-maf musculoaponeurotic fibrosarcoma oncogene homologue), promotes insulin gene transcription in response to increased blood glucose by binding to conserved cis-acting regulatory elements (A3, C1, E1 elements) present in the insulin promoter region [18,19,20,21]. Although MafA can activate the insulin promoter alone by binding to the C element, the tethering of the BETA2/NeuroD1 and PDX1 to the elements E and A [22] synergically increases the insulin transcription rate [23,24,25]. The stability of the MafA transcription factor and its binding to the regulatory elements of the *Ins* promoter is modulated by phosphorylation, in a glucose-dependent manner [26].

Once synthesized, insulin mRNA binds to specific regulatory elements that escort it to the ribosomes to initiate translation [27,28]. Newly synthesized pre-proinsulin (110 amino acids) contains a hydrophobic N-terminal sequence that recognizes cytosolic ribonucleoprotein signal recognition particles (SRPs) facilitating its translocation across the endoplasmic reticulum (ER) membrane [29,30]. This process, in which translation is coupled to translocation, was long considered to be the only mechanism responsible for the protein accumulation into the ER; about 5–15% of synthesized pre-proinsulin molecules are not directly translocated into the ER [31,32]. These molecules are released in the cytoplasm and rescued through the post-translation translocation route, which is SRP-independent and relies on the specific component of the translocation machinery, as Sec62/63 and the signal sequence receptor TRAP [32,33,34]. 

Upon delivery to the ER lumen, the signal peptide is rapidly cleaved to yield proinsulin, which undergoes chaperone-mediated folding forming three disulphide bonds. The ER folding machinery meticulously controls proinsulin in order to recognize misfolded or unfolded molecules that are eliminated through ERAD (ER-associated protein degradation)-dependent ubiquitination or ER-macroautophagy [31,35,36,37]. An accumulation of misfolded proinsulin activates the unfolded protein response (UPR) in order to alleviate the ER load and restore the folding capability [38]. UPR activation is primarily mediated by the heat shock protein/chaperone BIP (GRP78), which dissociates from and activates PERK (protein kinase R-like ER kinase), IRE1 (interferon response element 1) and ATF6 (activating transcription factor 6), the three major ER stress sensors [22,38,39]. Through the phosphorylation of eIF2α, PERK induces the transcription of CHOP (C/EBP homologous protein) and ATF4, which alleviates the ER load by reducing the pre-proinsulin translation; PERK can also support the expression of ATF6, another key component of the UPR machinery [40,41,42,43]. ATF6 translocates from the ER to Golgi, where it is cleaved, and migrates to the nucleus where it induces the transcription of CHOP, XBP1 and other chaperones involved in protein folding and degradation [44,45,46]. Protein folding is also supported by the activation of IRE1, which promotes the transcription of genes encoding for folding enzymes [43,47]. In healthy β-cells, approximately 20% of proinsulin misfold, resulting in a basal activation of the UPR machinery that continuously restores ER homeostasis. On the other hand, stressed β-cells (as seen in T2D) could exhibit severe ER homeostasis defects that may result in proinsulin misfolding [48]. When the ER stress levels increase beyond a threshold, either because of excessive protein synthesis demand (and overworked β-cells) or due to over accumulation of misfolded proteins in the ER lumen, the UPR sensors trigger signaling pathways and transcriptional events that induce β-cell dysfunction and death [38,49]. A very recent work has revealed that a pharmacological reversal of UPR or protein aggregation in β-cells improved cellular insulin content and release [50], reinforcing the ER homeostasis-to-insulin biosynthesis link in the pancreatic β-cells.

Folded proinsulin is delivered through COPII vesicles to the Golgi, where a complex set of events coordinates insulin granule formation and maturation [27]. Upon transport through the Golgi complex, proinsulin, together with its processive enzymes, is loaded into immature secretory granules, characterized by high Ca^2+^ levels and low pH, the optimal environment for the activation of endopeptidases. The proteolytical cleavage of proinsulin by prohormone convertase 1/3 (PC1/3) and the removal of the C-terminal domain by carboxypeptidase E (CPE) result in the formation of mature insulin, consisting of A and B chains held by two disulphide bonds [51,52,53]. The cleaved C-terminus, known as C-peptide, remains within the granules and is co-secreted with insulin. Along with the proinsulin-to-insulin conversion, the maturation of insulin granules requires a shift in proteins and lipid composition, which leads to the acquisition of competence for stimulus-dependent secretion [27,54]. During transport in the trans-Golgi network (TGN), granules expose their membrane adaptor proteins, a process that facilitates the assembly of the clathrin coat and its targeting to the plasma membrane [54,55]. Changes in lipid composition occur as well; indeed, membrane cholesterol progressively increases reaching 35 mol% in the TGN, a concentration similar to that found in the plasma membrane [56,57,58]. This cholesterol enrichment is required for the sorting of endopeptidases and adaptor proteins within the forming granules [9] and to confer the membrane curvature necessary for the budding of granules from the TGN [11]. Then granules progressively mature and acquire a dense core formed by Zn^2+^–insulin crystals, while unwanted cargoes and membrane proteins are retrieved back to the TGN by retrograde transport [59].

Once mature, insulin granules are distributed into two different pools, a small fraction (1–5%), known as the ready releasable pool (RRP), is docked to the plasma membrane and is released on demand without any further modification, while the remaining granules (95–99%) belong to the reserve pool (RP), which must undergo a series of preparatory reactions before acquiring release competence [16,60]. 

### 2.2. Insulin Secretion

Insulin secretion shows a canonical biphasic pattern consisting of a rapid and transient first phase due to the release of RRP granules, followed by a prolonged second phase [60,61,62]. The first phase begins when blood glucose levels exceed 100 mg/dL, which leads to its internalization through GLUT1 transporters (GLUT2 in rodents) into the β-cells [63,64]. Once in the cytoplasm, glucose is phosphorylated by the glucokinase (GCK), enzyme that functions as a glucose sensor [8,65]. At low intracellular glucose levels, GCK is inactive and associated with insulin granules through the interaction with nNOS (dimeric neuronal nitric oxide synthase). The increase of intracellular glucose promotes the dissociation of GCK from nNOS and its translocation into the cytoplasm, where GCK phosphorylates glucose to generate glucose-6-phosphate that enters into the glycolytic pathway [16]. Pyruvate, the final product of glycolysis, is then oxidized via the Krebs cycle into the mitochondria to produce ATP. The increased ATP/ADP ratio induces the closure of K^+^_ATP_ channels leading to cellular depolarization, which in turn, triggers the opening of L-type voltage gated Ca^2+^ channels (VDCC) [66,67]. The rapid calcium influx promotes the fusion of insulin granules with the plasma membrane, a process that involves the spatial reorganization of multiprotein machinery known as the SNARE (soluble-N-ethyl-maleimide sensitive factor-attachment protein receptor) complex. In mature pancreatic β-cells, the main components of SNAREs are syntaxin 1 and SNAP-25 (soluble NSF (N-ethylmaleimide-sensitive factor) attachment protein receptor) on the plasma membrane and the vesicular protein VAMP2 (vesicle-associated membrane protein 2), also known as synaptobrevin [68]. The SNARE complex acts as a zipper, tethering the granules to the plasma membrane, and ensuring Ca^2+^ entry in the proximity of docked granules. Indeed, syntaxin1 and SNAP-25 bind the second and third intracellular loops of the L-type calcium channels, keeping them in close contact with the secretory granules [16,61,68]. The spatial organization of the SNARE complex is guaranteed by their localization in specific domains within the plasma membrane, known as lipid rafts, where L-type Ca^2+^ and K^+^_ATP_ channels are also enriched [69]. Alongside SNAREs, Synaptotagmin (Syt) family proteins play a versatile role in insulin exocytosis. While Syt-7 has been recognized as aiding granule recruitment to the plasma membrane upon glucose stimulus [70], Syt-4 has been indicated to regulate the RRP insulin granule quantities as well as the localization of the granules to the plasma membrane [71]. The second phase of insulin secretion relies on the mobilization of granules belonging to the reserve pool (RP) and can be maintained for several hours until glucose levels remain elevated [62]. Unlike the first phase, prolonged insulin release is mainly stimulated by metabolic intermediates, as acetyl-CoA, ATP, NADPH, and α-ketoglutarate/glutamate, that amplify the glucose-induced insulin secretion [72]. Besides glucose, other substrates including amino acids, lipids and paracrine signals can control insulin secretion using parallel mechanisms [73]. 

## 3. Cholesterol Biosynthesis, Homeostasis and Turnover 

Insulin biosynthesis, sorting and secretion are affected by cholesterol levels as the lipid not only represents a fuel for the β-cells but also a key constituent of their membrane domains. Thanks to its rigid and planar structure, cholesterol contributes to the physical properties and organization of cellular membranes affecting cellular permeability, signaling activity, organelles architecture and vesicular trafficking [12,74]. Given the key structural and functional roles of cholesterol, pancreatic β-cells are equipped with sophisticated machinery able to control cholesterol synthesis, uptake and efflux, as well as its intracellular distribution [8] (Figure 2). 

### 3.1. Cholesterol Biosynthesis

Pancreatic β-cells synthetize cholesterol in the ER through the mevalonate pathway, a complex cascade of enzymatic reactions starting with the condensation of acetyl-coenzyme A (CoA) and acetoacetyl-CoA to form 3-hydroxy-3-methylglutaryl-CoA (HMGCoA). This reaction is catalyzed by hydroxymethylglutaryl-CoA reductase (HMGCoAR), which represents the limiting enzyme for cholesterol synthesis [76]. The transcription of HMGCoAR is controlled by the transcription factor SREBP2 (sterol-responsive element binding protein 2), which is active only when cholesterol levels are low. SREBP2 is present in the inactive form, bound to the SREBP cleavage-activating protein (SCAP), which is the real sensor of sterols in the cell. When cholesterol concentration decreases, the SREBP2 detaches from SCAP and translocates into the nucleus where it binds to sterol regulatory sequences (SREs) present in the HMGCoAR promoter region, favoring its transcription [8,77]. The importance of SREBP2-HMAGCoAR axis in β-cell homeostasis was demonstrated by studies in transgenic mice overexpressing SREBP2. Cholesterol accumulation in the transgenic model of in vivo activation of SREBP2 resulted in reduced β-cell mass and insulin-deficient diabetes [77]. 

### 3.2. Cholesterol Uptake

Cholesterol uptake from plasma lipoproteins represents another mechanism for cholesterol accumulation into β-cells. Lipoprotein transport is mediated by receptor systems that selectively recognize and bind LDL (low density lipoprotein), VLDL (very low density lipoprotein) and HDL (high density lipoprotein). Pancreatic β-cells are equipped with different lipoprotein receptors, including the LDL receptor (LDLR), the VLDL receptor (VLDLR), the receptor for oxidized LDL (CD36) and the scavenger B1 receptor (SR-B1), which is selective for HDL [12,78]. Unlike the others, LDLR requires co-receptors for its function, as LPR5; the importance of LPR5 for LDLR activity has been demonstrated in LPR5 knock-out mice, which showed reduced hepatic chylomicron clearance, insulin secretion and glucose tolerance [79]. Moreover, global LDLR-KO mice displayed increased blood cholesterol levels and were susceptible to high-fat diet with purified islets from those showing poor glucose-stimulated insulin secretion [80]. Likewise, a study using β-cell-specific knockout of LDL receptor-related protein 1 (LRP1) explored the relationships between the type of diet, the LDL receptor function and lipid metabolism [81]. The majority of studies regarding the impact of dysregulated cholesterol uptake on insulin secretion have been focused on LDLR as it is highly expressed in pancreatic β-cells, as shown in Figure 2. 

Cholesterol uptake is directly regulated by cholesterol availability via modulation of LDLR transcription, a process mediated by SREBP2, which therefore represents the master regulator of cholesterol levels in β-cells [77]. An alternative mechanism is represented by post-translational regulations of LDLR expression. In the liver, LDLR abundance on the plasma membrane is controlled by the proprotein convertase subtilisin/kexin type 9 (PCSK9), an enzyme that routes LDLR to the lysosomes for degradation, thus preventing its recycling to the plasma membrane [82,83]. Very recently, we and others have shown that LDLR expression, and thus cholesterol uptake, are modulated by PCSK9 also in pancreatic β-cells [84,85,86,87]. PCSK9 full knock-out mice exhibit reduced plasma cholesterol levels, but increased cholesterol content in the islets of Langerhans, as a consequence of increased LDLR expression in β-cells. Mice developed glucose-intolerance that appears to be the result of insulin insufficiency rather than insulin resistance. In accordance, PCSK9 KO mice presented altered islets morphology, along with dysregulated insulin release [85,87]. In contrast, mice in which PCSK9 is switched off selectively in the liver exhibited intact glucose metabolism, revealing that pancreatic PCSK9, rather than the hepatic-circulating protein, is involved in the control of LDLR expression in pancreatic islets [8,85]. Indeed, PCSK9 can be synthesized and secreted by pancreatic δ- and β-cells and might act locally in paracrine/autocrine manners. We have recently confirmed this hypothesis in a novel animal model where PCSK9 expression was deleted only in PDX1 expressing cells, namely β- and δ-cells; transgenic mice exhibited cholesterol accumulation in the β-cells and defective insulin secretion, confirming that the pancreatic PCSK9-LDLR-cholesterol axis plays a prominent role in the regulation of cholesterol homeostasis in pancreatic islets [86]. In contrast to these findings, a separate study involving gene silencing or pharmacological inhibition of PCSK9 in human EndoC-βH1 cells suggested that while PCSK9 knockdown or functional inhibition resulted in increased expression of cell surface LDLR, glucose stimulated insulin secretion decrement was not observed, however observing ~50% loss in GSIS when cells were directly exposed to LDL [88]. Species difference (human vs. mouse) or the stage of β-cell differentiation (embryonic vs. adult) could be likely explanations for the observed phenotype differences. In accordance, expression analysis of cholesterol handling proteins highlighted a great heterogeneity among β-cell lines; thus, it is also possible that distinct compensatory mechanisms in β-cells might set in upon the severe loss of PCSK9. Further investigations are necessary to clarify this point. 

### 3.3. Cholesterol Esterification and Efflux

As excess free cholesterol can be toxic, cells normally convert sterols in cholesteryl esters by the enzyme acyl-coenzyme A:cholesterol acyltransferase (ACAT) 1 and store them in lipid droplets. Little is known about ACAT1 in β-cells, but lipid droplets have been observed in human islets of Langerhans and found to be enriched in aged and T2D islets [89]. An additional mechanism to remove excess cholesterol is the efflux via the SR-B1 receptor (which mediates both cholesterol uptake and efflux) and ATP-binding cassette (ABC) transporters, which transport cholesterol against gradient through ATP hydrolysis [12]. ABC transporters belong to five different subfamilies (A–G), which have different cellular distribution and substrate specificity [90]. Among them, ABCA1 and ABCG1 are the main players in mediating cholesterol efflux, as suggested by the fact that genetic mutations of ABCA1 lead to the Tangier disease, a genetic disorder characterized by cholesterol accumulation in various organs, atherosclerosis and premature coronaropathy [91,92]. ABCA1 transports cholesterol specifically to HDL lipoproteins, while ABCG1 mediates cholesterol efflux to non-specific acceptors as HDL, LDL and cyclodextrin [90]. As in other tissues, ABCA1 and ABCG1 are the major contributors of cholesterol efflux also in pancreatic β-cells as revealed by the increased cholesterol accumulation in pancreatic islets of ABCA1 and ABCG1 knock-out mice [93,94]. The expression of ABCA1 and ABCG1 is strictly regulated by the intracellular content of cholesterol and/or oxysterols (oxygenated derivatives of cholesterol). Indeed, these sterols promote the transcription of ABCA1 and ABCG1 functioning as ligand-activators of the LXR-RXR pathway (liver-X-receptor/retinoic acid receptor) [95]. Accordingly, disruption of the LXR-RXR axis, due to genetic ablation of LXR, causes increased cholesterol accumulation and β-cell dysfunction [96]. 

## 4. Cholesterol Distribution and Trafficking

Intracellular cholesterol levels are fine-tuned by the two transcription factors SREBP2 and LXR. Under cholesterol depletion, SREBP2 is activated and controls the transcription of both HMGCoAR and LDLR to ensure the rapid sterol increase via de novo synthesis and uptake from the extracellular milieu. On the other hand, when cholesterol increases, LXR prevails and controls the expression of ABC transporters to prevent the toxic accumulation of the lipid. The critical role of these pathways in insulin secretion and glucose homeostasis has been largely confirmed by knock-out experiments in animals and by genetic studies in humans [8,77,96]. 

A detailed analysis of intracellular membrane composition reveals that β-cells not only control the total cholesterol content but also its redistribution within intracellular organelles. A vast majority of cholesterol resides in the plasma and insulin granules membranes (60–90%), while only a small amount can be found in the ER (0.5–1%). Intermediate levels can be detected in the Golgi apparatus, mitochondria, lysosomes and peroxisomes [97]. This precise cholesterol distribution is essential for the optimal functioning of these compartments; however, the mechanisms by which specific membrane lipid compositions are established and maintained are poorly understood in β-cells. As the sterol is synthesized in the ER or accumulated in lysosomal compartments by uptake systems, the main challenge is to understand the mechanism by which this hydrophobic, poorly water-soluble lipid molecule is transported between organelles through the aqueous environment of the cytosol to reach the functional cholesterol concentration. Figure 3 summarizes the possible intracellular itinerary of cholesterol in pancreatic β-cells.

### 4.1. Mechanisms of Intracellular Cholesterol Trafficking

Potential mechanisms for intraorganellar trafficking of cholesterol, as well as other lipids, are vesicular and non-vesicular transport pathways. The vesicular-mediated transport involves the incorporation of cholesterol into the membranes of vesicles/granules that move along the actin cytoskeleton to reach the intracellular compartments where cholesterol is required. This transport is energetically expensive; therefore, cells have evolved alternative mechanisms for cholesterol delivery [98,99]. Among them is the non-vesicular transport via soluble cytosolic lipid transfer proteins, called STPs (sterol transfer proteins). Although several candidate STPs have been identified and shown to be able to mediate sterol transfer in vitro [100,101], assigning a clear physiologic function in vivo to them has proven challenging. STPs are divided in two classes: the steroidogenic acute regulatory protein (StAR)-related lipid transfer (START) domain proteins (STARDs) and the oxysterol-binding proteins (OSBPs)/OSBP-related proteins (ORPs) [98,102]. In mammals, the STARD family consists of 15 members that harbor a highly conserved C-terminal sequence (210 amino acids) and an α-helix domain. The C-terminal represents the binding pocket, while the α-helix operates as a lid that modulates lipid entrance and egress [103]. STARD proteins bind and deliver sterols, phospholipids and sphingolipids; cholesterol transport is mainly mediated by STARD1, STARD3 and STARD4 subfamilies [104,105]. The second group of STPs includes the OSBPs/ORPs family consisting of 12 members in mammals [106,107]. ORPs possess a highly conserved C-terminal OSBP homology domain (ORD, 400 amino acids) that binds cholesterol, ergosterol, oxysterols and other lipids, as phosphatidylinositol-4-phosphate and phosphatidylserine. The majority of ORPs also possess a pleckstrin homology domain and a di-phenylalanine in the FFAT motif of the N-terminal sequence useful for membrane recognition and binding [108]. Interestingly, STPs are not merely cholesterol transporters but can act also as sensors that regulate the expression of genes encoding for proteins involved in cholesterol synthesis (SREBP2 and HMGCoAR) and metabolism (ACAT1 and ACTA2) [109,110]. STPs of both groups are expressed in β-cells (Figure 2), but very little is known about their role in β-cell pathophysiology [111,112,113]. 

A third potential mechanism for non-vesicular transport is via the formation of membrane contact sites (MCs) where cholesterol can be exchanged between the organelles without transiting in the cytosol. MCs are widespread in eukaryotic cells, can occur between all membranes and are extremely dynamic as organelles must rapidly associate and dissociate to respond to the metabolic needs of the cell [114]. The organelles involved in MCs are tethered (not fused) at a distance of 10–80 nm via multiprotein complexes, comprising structural, functional and regulatory proteins. MCs present peculiar proteomic and lipidomic profiles according to their physiological role [115]; for instance, mitochondrial-ER contact sites, also known as MAM (mitochondrial-associated membranes), are enriched in proteins that ensure the transport of phosphatidylserine (PS) from the ER to the mitochondria, where the phospholipid is converted into phosphatidylethanolamine, essential for mitochondrial function [116]. Even though this transport has not been identified yet in pancreatic β-cells, several lines of evidence recently suggested that MAMs occur also in these cells and are involved in the control of insulin secretion [117,118]. Whether they also play a role in the intracellular redistribution of cholesterol is not known, but interestingly, alteration of their organization has been detected in T2D and T1D scenarios [119,120,121,122]. Furthermore, palmitate treatment of Min6-B1 cells resulted in altered GSIS, elevated ER stress and decreased ER-mitochondrial contacts, potentially linking interorganellar interactions, insulin secretion and lipotoxicity [119]. Fatty acid-mediated ER damage and cholesterol dysregulation in β-cells are concisely highlighted later in this review (see Section 5.1).

### 4.2. Intracellular Itinerary of Cholesterol 

Another open question is how cholesterol is sensed in the different intracellular compartments and how its precise concentration is maintained. ER and lysosomes are emerging as critical nodes in cholesterol homeostasis and redistribution [99,123]. ER represents the site of cholesterol synthesis, which is controlled by resident sterol sensors able to regulate the sterol biosynthesis or efflux via translational control. In the ER, cholesterol concentration must be tightly regulated (not exceed 0.5–1%) for an efficient control of mRNA transduction and protein folding [32,99]; therefore, the neosynthesized lipid must be directed toward different organelles. The expected route of cholesterol redistribution is toward the Golgi apparatus, where its concentration progressively increases moving from the cis- to the trans-side. This forward cholesterol trafficking is absolutely required to sustain the continuous granules generation at the TGN. This route has been recently investigated in β-cells and found to involve the oxysterol binding protein OSBP, which mediates the non-vesicular transport of cholesterol between the ER and the TGN [113]. Hussain and co-workers demonstrated that loss of OSBP reduces the stability of newly formed insulin granules and impairs proinsulin synthesis, affecting the ability of β-cells to secrete the proper amount of the hormone [113]. 

An alternative route for cholesterol is from the ER to the plasma membrane (PM) where the lipid participates in the organization of rafts, cholesterol-enriched membrane domains important for the recruitment of the signaling machinery necessary for the stimulus-secretion coupling in β-cells. CAV1 (caveolin 1) has been proposed as an important cholesterol sensor in the ER since it delivers cholesterol to the PM, when its concentration is high. The key role of CAV1 was demonstrated by increased cholesterol accumulation and decreased stability of MAMs in embryonic fibroblasts of CAV1-deficient mice [124]. Interestingly, CAV1 is also expressed by β-cells, in particular in membrane rafts and insulin granules, and mutations in the gene are associated with the development of T2D [125]. In vitro studies have shown that CAV1 controls insulin secretion by forming complexes with the insulin granule proteins VAMP2, cdc42 (Rho GTPase cell division cycle 42) and βPIX (guanine nucleotide exchange factor 7) [126]. In line with these findings, CAV1 silencing/ablation in β-cell lines or mice have been associated with hyperinsulinemia under physiological lipid and glucose levels [127,128]. Finally, a possibility is also cholesterol transfer between ER and mitochondria, a process mediated by the TSPO/VDAC/STARD1/ANT complex at MAMs, although this route is unlikely in physiological conditions, as both these organelles require low cholesterol levels for their optimal functioning [129]. 

In recent years, the lysosomes have been proposed to play a key role in cholesterol homeostasis [130]. Circulating LDL-cholesterol is internalized in pancreatic β-cells by receptor-mediated endocytosis (LDLR) and delivered to the lysosomes via clathrin-coated vesicles, where LDL-cholesterol is metabolized by acid lipases. From lysosomes, unesterified cholesterol is transported to downstream organelles to reach the optimal concentration [131,132,133]. The release of cholesterol from lysosomes is normally mediated by NPC1 (Nieman–Pick type C1) and NPC2 (Nieman–Pick type C2) proteins [134,135,136]. NPC2 tethers the hydrophobic portion of cholesterol, keeping it close to the N-terminal domain of NPC1, which by binding to the 3β-hydroxyl group, embeds cholesterol into the lysosomal membrane. Different proteins interact with NPC1 to accept cholesterol, ensuring its delivery to different intracellular compartments [136,137,138]. NPC1 not only functions as a transporter but also as a sensor of sterol levels. Indeed, cholesterol overload in lysosomes causes both the NPC1-mediated LXR activation to ensure the upregulation of cholesterol efflux systems and the NPC1-mediated SREBP2-HMGCoAR repression to prevent its de novo synthesis [139]. NPC1 also modulates the activity of mTORC1 (mammalian target of rapamycin complex 1), a key protein involved in the sterol biogenesis through regulation of SREBP1c and SREBP2 expression [140,141], and control of the autophagic flux [142]. The importance of proper cholesterol efflux from the lysosomes and the relevance of NPC1/2 proteins in the process are highlighted by the Niemann–Pick disease, in which NPC1/2 inactivating mutations cause the abnormal accumulation of free cholesterol in the lysosomes of almost all cells, especially neurons and hepatocytes [134,143]. 

β-cells express both NPC1 and NPC2; however, how cholesterol is transported to downstream organelles is completely unknown. Studies from other cells indicate that about 30% of lysosomal cholesterol moves toward the ER, where it is integrated with the neo-synthesized pool [99]. Cholesterol can be transported directly to the ER through membrane contact sites between the two organelles, tethered by the ORP1L/ORP5/VAP/STARD3 bridging complex as observed in Hela and MelJuso cells [144,145]. This mechanism, however, seems unlikely under physiological conditions, as high lysosomal cholesterol levels destabilize the interaction between tethering proteins [99]. Recent studies in CHO-K1 cells suggest that lysosomal cholesterol accumulates in the plasma membrane, before reaching the ER [146]. Even though the mechanisms responsible for cholesterol transport from the lysosomes to the plasma membrane are not fully understood, it has been proposed that NPC1 promotes the formation of cholesterol-enriched lysosome-related organelles (LROs), which move along the cytoskeletal tracks to reach the plasma membrane (PM) [99,102]. Once in the plasma membrane, cholesterol distributes within the phospholipid bilayer and assemblies with multiple proteins to form the lipid rafts. Excessive cholesterol at the plasma membrane must be removed and delivered to the ER via a retrograde transport that is mediated by STARD4-, ORP1- and ORP2-dependent pathways as shown in primary hepatocytes and HepG2 cells [147,148]. The importance of STPs in modulating cholesterol retrograde flow has been demonstrated by the accumulation of cholesterol in the plasma membrane of STARD4-deficient cells [147]. Another possible mechanism is the formation of membrane contact sites between the ER and the plasma membrane mediated by the interaction between phosphatidylserine (PS) and Aster proteins, expressed in both organelles. As shown by Trinh and co-workers, when PM cholesterol increases in CHO-K1, Aster protein binds PS forming a bridge between ER and PM allowing cholesterol delivery [146]. This retrograde flow might be particularly relevant in pancreatic β-cells under starving conditions; however, the underlying mechanisms are completely unknown in these cells. 

Direct cholesterol redistribution from the lysosomes to other organelles has been proposed. For instance, NPC1- and VAMP4-vesicular mediated transports have been proposed between the lysosomes and the Golgi apparatus in NRK and CHO cells [149,150]. Cholesterol can also be delivered directly to the mitochondria via a STARD3-dependent mechanism, as suggested by a significant reduction of mitochondrial cholesterol in STARD3-deficient COS1 cells [151]. Furthermore, lysosomes can form dynamic and extensive membrane contact sites with peroxisomes, which are mainly mediated by the tethering between lysosomal synaptotagmin VII and peroxisomal PIP2 (phosphatidylinositol 4,5-bisphosphate). Probably, the close juxtaposition between the two organelles facilitates the efficient cholesterol transport via STARD and/or ORPs [152]. Excessive cholesterol can be also stored in lipid droplets, which exchange cholesterol with almost all organelles via the formation of membrane contact sites [89].

Most of the proteins involved in intracellular cholesterol redistribution are expressed by human and clonal β-cells (Figure 2), thus suggesting that these pathways may potentially occur in these cells; however, their relevance to β-cell pathophysiology remains to be investigated.

## 5. Effects of Cholesterol Imbalance on the Insulin Biosynthetic and Secretive Pathway

While the detailed description of the cholesterol flux within the β-cell is still an open question, several lines of evidence revealed that the perturbation of cholesterol homeostasis (excess or depletion) impairs the insulin secretion through different mechanisms. 

The impact of cholesterol overload in the latest stages of insulin exocytosis has been extensively studied [9,10,93]; however, less is known about its impact on insulin transcription, synthesis and granulogenesis. In the following section, we will briefly summarize how cholesterol imbalance could impair (1) ER homeostasis, (2) insulin biosynthesis and granules maturation and (3) insulin secretion. The main effects of cholesterol overload on pancreatic β-cells are reported in Figure 4.

### 5.1. Impaired Cholesterol Regulation Affects β-Cell ER Homeostasis

Chronic exposure of β-cell to fatty acids leads to a condition called lipotoxicity that in some way reflects events featured in obesity and T2D [153]. β-cell line models have been extensively utilized to study effects of lipotoxicity, and interesting observations have been made in β-cells that include ER stress and apoptosis. With regards to the topic under discussion, β-cells treated with palmitate for prolonged time (48 h) resulted in a decrease of ER sphingomyelin and cholesterol, thereby disrupting the ER lipid rafts, which led to ER protein trafficking defects [154], suggesting that cholesterol-mediated stability of ER lipid drafts could contribute to ER proteostasis. Independent of this study, how palmitic acid exposure negatively affects GSIS [155,156], which could partially be a result of perturbed proinsulin trafficking in β-cells, has been reported by many. Similar to the effect of decreased cholesterol, too much of it could be detrimental too, just like described earlier in β-cell PCSK9 KO model. Direct cholesterol loading of β-cells (a situation mimicking hypercholesterolemia) has shown ER stress, apoptosis and β-cell dysfunction [157,158]. From a pathophysiology perspective, under LDL toxicity, intracellular LDL lipid oxidation could produce cholesterol peroxides and β-cell oxidative stress due to a build-up of reactive oxygen species (ROS) and lipid peroxides [159]. Oxidative stress has been indicated as a contributing factor to ER stress and insulin secretory impairments in a separate study where β-cells were administered with mildly oxidized LDL [160]. Excessive cholesterol in the ER has been proposed to inhibit the SERCA2b (an ER calcium pump) activity in non-β cells [161]. With recent evidences pointing out that a loss of SERCA2 could disrupt ER protein trafficking and proinsulin processing in β-cells [162], it can be envisaged that cholesterol enrichment in β-cells could impede anterograde transport of ER proteins including proinsulin by affecting SERCA2 functions. It is yet to be understood if proinsulin folding in the ER, a necessary step for insulin biosynthesis, is affected by the aforementioned impediments. Overall, maintenance of an optimum level of cholesterol in the ER is vital for its structure and proper functioning.

### 5.2. Cholesterol Imbalance, Insulin Biogenesis and Granules Formation

Insulin biogenesis begins in the ER, the organelle also responsible for sorting newly synthesized and exogenous cholesterol toward different cellular compartments [113]. The increased flux of cholesterol places a tremendous burden on the ER that must redistribute the sterol in other organelles to maintain its optimal function. The first option is to deliver cholesterol to the plasma membrane in a CAV1-dependent manner but, when its concentration increases beyond threshold, it may return to the ER via a retrograde transport. Excessive cholesterol can then be delivered to the mitochondria. The increased cholesterol flow into the mitochondria results in increased membrane potential and ROS production, which impair the ATP production [163]. Increased oxidative stress, which is exceptionally dangerous in β-cells due to the low availability of antioxidant enzymes, downregulates the expression and activity of PDX1 and MafA, affecting the insulin gene transcription [164,165]. Elevated ROS also trigger the UPR, which affects the ability of the ER to cope with misfolded/unfolded proinsulin. The progressive sequestration of chaperones to unfolded proinsulin downregulates the ERAD-mediated degradation of misfolded proteins resulting in ER overload of misfolded proteins like proinsulin [38]. Therefore, the net effect of cholesterol accumulation in the ER is the reduction of well-folded proinsulin trafficking to the Golgi apparatus, compromising the ability of β-cells to manufacture, process and store insulin.

Excessive cholesterol in the ER can also be delivered to the Golgi complex, where it accumulates in the cisternae, inducing their spatial reorganization. In accordance, cholesterol overload due to the ablation of ABCA1 expression alters the Golgi ultrastructure, inducing the stacking of Golgi cisternae and the reorganization of Golgi ribbons into circular structures [13,94]. This spatial rearrangement has been observed to block the forward protein delivery out from the Golgi apparatus in kidney cells. Similarly, Golgi deformation in β-cells by cholesterol can affect (a) the post-ER proinsulin transport and localization (proinsulin normally is localized in a juxtanuclear Golgi region [166]); (b) subcellular localization and function of prohormone convertases; and, finally, (c) impairment of the insulin granules packaging and trafficking toward the plasma membrane, independent of its effect on proinsulin and processing enzymes.

Excessive cholesterol also accumulates directly into the insulin granules, which must maintain optimal cholesterol concentration for their proper function. Increased cholesterol may affect the biophysical properties of the membrane bilayer, altering the granule membranes curvature and leading to the activation of phospholipases, which control the membrane lipid composition. For instance, cholesterol overload has been shown to cause the recruitment of phospholipase A2 (cPLA2) to the insulin granules, which converts phospholipids into arachidonic acid and lysophospholipids, changing the topological constraints on the membranes [167]. This lipid reorganization, in turn, may alter the sorting of cargos, coat adaptor proteins and SNAREs proteins important for the correct maturation and trafficking of granules to the plasma membrane or for the maintenance of the TGN structure. In accordance, cholesterol overload in INS1 and MIN6 cells results in insulin granules enlargements and altered membrane curvature [11]. These alterations cause the accumulation of immature insulin granules that leads to a massive “traffic jam” in the Golgi complex, reducing the ability of β-cells to secrete the proper amount of insulin [9,11]. In line with these observations, we have recently showed that LDL-cholesterol overload, due to PCSK9 silencing, strongly increases the proinsulin/insulin ratio in mouse β-cells, revealing a potential role of the LDLR-PCSK9-cholesterol axis in the regulation of granules formation and maturation [86]. Since granule maturation is paralleled by progressive cholesterol incorporation into their membranes, cholesterol depletion can be deleterious as well. In accordance, treatments with lovastatin, an inhibitor of HMGCoAR, lead to insulin granules enlargement and proinsulin accumulation, reducing the ability of β-cells to secrete insulin [168]. In recent years, it has been emerging a potential role of ABCG1 in maintaining insulin granules integrity by facilitating cholesterol accumulation within granules membrane. An elegant study by Hussain and co-workers demonstrated that ABCG1 depletion strongly reduces the cholesterol content in granule membranes, making them unable to fully condense and pack insulin molecules. ABCG1 downregulation also decreases the total number of insulin granules, suggesting a potential elimination of immature insulin granules through endosome-dependent and independent pathways [113].

### 5.3. Dysregulated Cholesterol Homeostasis and Insulin Secretion

Cholesterol accumulation dampens the glucose-stimulated insulin secretion at an early stage by altering the ability of pancreatic β-cells to transport glucose, as revealed by the reduction of glucose transporters activity (50% decrease) in mouse β-cells exposed to high cholesterol [163]. Cholesterol also modulates the rate of the glycolytic pathway by regulating the dimer-to-monomer ratio of nNOS, which controls GCK availability into the cytosol. Hao and co-workers demonstrated that cholesterol accumulation stabilizes the dimeric form of nNOS, preventing its translocation from insulin granules to the cytosol, which strongly reduces the glycolytic flux and the ATP production [10].

Another critical step in the insulin secretive pathway is the generation of ATP into the mitochondria. Cholesterol overload alters mitochondrial membrane fluidity and permeability leading to a redistribution of the electron transport chain complexes and a reduction of the proton gradient, which in turn strongly reduces the ATP production and the insulin release [169]. Cholesterol might also reduce the mitochondrial function by altering MAMs organization and stability. Indeed, it potentially affects the stability and extension of MAMs, which regulate Ca^2+^ exchange between mitochondria and ER, as well as mitochondrial dynamics [117,118]. The latter process is crucial for maintaining an efficient pool of mitochondria as it regulates their morphology and allows the elimination of dysfunctional organelles [170]. Furthermore, cholesterol regulates the distribution of key enzymes involved in the replication and transcription of the mitochondrial DNA (mtDNA). Among them, mtDNA polymerase polG and mtDNA helicase Twinkle that are associated with cholesterol-enriched membrane platforms close to the ER. A recent study showed that cholesterol accumulation altered the localization of these enzymes in the mitochondrial membrane leading to mtDNA aggregation and reduction of its synthesis. These alterations might affect the generation of mtDNA-encoded mitochondrial complexes reducing the ability of mitochondria to produce ATP [171].

Cholesterol also controls the docking and fusion of insulin granules with the plasma membrane by regulating the formation of the lipid rafts and, thus, the organization of the SNARE complex. Disruption of cholesterol efflux due to ABCA1 silencing in β-cells leads to cholesterol overload in membrane rafts, which is associated with impaired glucose- and KCl-stimulated insulin secretion [13,172]. In line with this observation, the knockout of LXRβ, which controls ABCA1 expression, causes β-cell dysfunction and decreases insulin secretion [96]. Likewise, LXR activation (by agonist T0901317) in MIN6 mouse insulinoma cells caused increased ABCA1 expression [173]. The role of LDLR in regulating insulin exocytosis is controversial as some studies observed cholesterol reduction and impaired insulin secretion in LDLR knock-out mice [174], whereas others reported no significant alterations in islet cholesterol levels nor β-cell dysfunction in these mice [172]. Recently, we have demonstrated that LDL-cholesterol accumulation, due to PCSK9 silencing, promotes the redistribution of the lipid rafts and reduces the expression of SNAP25 and VAMP2. These alterations cause the reduction of insulin granules docked to the plasma membrane, already in resting condition, leading to a significant decrease of both glucose- and KCl-stimulated insulin secretion [86]. Cholesterol depletion affects insulin granules exocytosis as well. As lipid rafts are cholesterol-enriched hotspots, cholesterol depletion reduces their stability leading to a redistribution of SNARE proteins and channels out of these microdomains [8,9]. In particular, cholesterol depletion induces the relocalization of syntaxin1 and SNAP25 from the lipid rafts, reducing the number of granules docked to the plasma membrane [175,176].

## 6. Conclusions

Cholesterol, as an essential component of the cellular membranes, exerts multiple structural and functional roles, orchestrating a wide range of signaling pathways and cellular functions. Cholesterol homeostasis is particularly relevant for pancreatic β-cells as they rely on this sterol for their survival and proliferation and for their functional maturation. It is clear that cholesterol influences β-cell ER structure and function, insulin biogenesis and exocytosis. It is therefore not surprising that cholesterol imbalance has been associated with the onset of multiple metabolic disorders, including T2D. To maintain cholesterol homeostasis β-cells have evolved sophisticated machinery based not only on the control of the lipid biosynthesis/conversion or influx/efflux but also on its dynamic redistribution within the different organelles. In accordance, it is emerging that cholesterol distribution, rather than its total level, might be of pivotal importance for the β-cell function. The mechanisms by which cells sense and shuttle the lipids in the different membrane compartments are still largely unexplored. Several sterol binding proteins identified in other cell types are also expressed by β-cells; therefore, assigning a clear physiological function to them may provide interesting new targets of intervention to control β-cell functionality. Indeed, increased intracellular cholesterol places a tremendous burden to the β-cell that becomes unable to properly deliver the sterol to its final destinations, resulting in altered plasma membrane organization, impaired mitochondrial activity, reduced insulin granules maturation and sorting. The net effect of this dysregulated environment is the inability of β-cells to manufacture, process and secrete insulin and β-cell dysfunction and/or death, which are hallmarks of T2D pathogenesis and progression. Therefore, a better understanding of the pathways regulating cholesterol itinerary and the functional consequences of their dysregulation might provide further insight of the molecular mechanisms involved in β-cell dysfunction and might be useful for the design of novel and improved pharmacological options for T2D treatment.

## Figures and Tables

**Figure 1 biomolecules-13-00224-f001:**
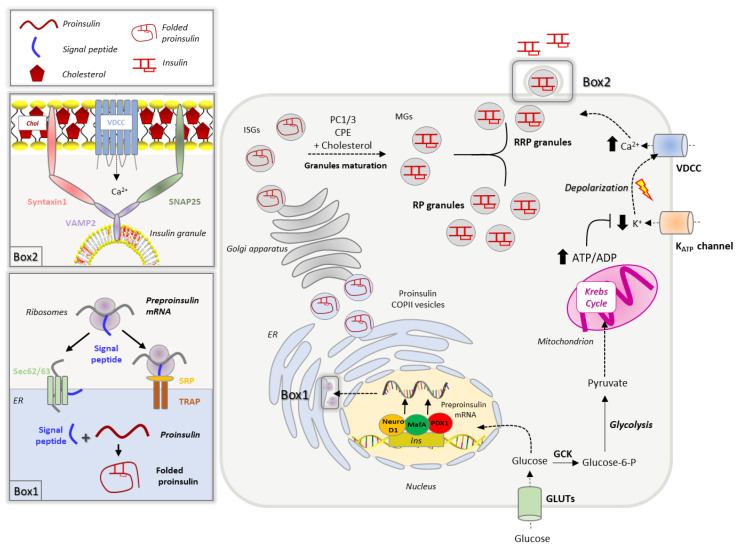
Insulin biosynthesis and secretion. Extracellular glucose is transported in the β-cells through GLUT transporters (GLUT1 in humans, GLUT2 in rodents). Glucose increase promotes the insulin gene transcription, which is controlled by PDX1, MafA and NeuroD1 transcription factors. The insulin transcript crosses the ER membrane via co-translational (SRP and TRAP-dependent) and post-translational (Sec62/63-dependent) translocation processes. Once in the ER lumen, the pre-proinsulin signal peptide is rapidly cleaved to yield proinsulin that undergoes oxidative folding (**Box 1**). Newly synthesized proinsulin, loaded into COPII vesicles, is then delivered to the Golgi apparatus, where it is sorted to immature secretory granules (ISGs). Granules maturation is associated with increased cholesterol accumulation in the membrane and proinsulin-to-insulin conversion mediated by prohormone convertases 1/3 (PC1/3) and carboxypeptidase E (CPE). Mature granules (MGs) are sorted into two different pools: the readily releasable pool (RRP) docked to the plasma membrane and the reserve pool (RP). Insulin secretion is promoted by the GCK-mediated phosphorylation of glucose, which is then metabolized via glycolysis and Krebs cycle, leading to an increase of ATP production. The elevation of ATP/ADP ratio promotes the closure of K^+^_ATP_ channels, membrane depolarization and activation of voltage-gated Ca^2+^ channels (VDCC). The increase of intracellular Ca^2+^ triggers the insulin granule fusion with the plasma membrane, a process regulated by the SNARE complex, composed by syntaxin1 and SNAP25 on the plasma membrane and the vesicular protein VAMP2; these multiprotein complexes occur in specific plasma membrane domains enriched in cholesterol (Chol), called lipid rafts (**Box 2**).

**Figure 2 biomolecules-13-00224-f002:**
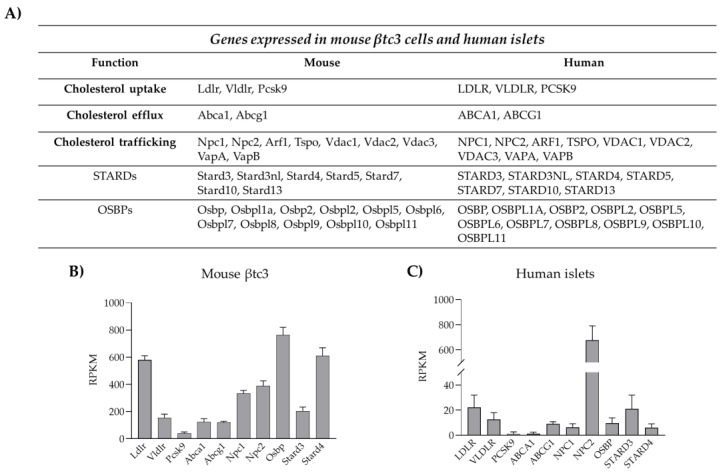
Machinery involved in cholesterol homeostasis and trafficking in pancreatic β-cells. (**A**) Genes involved in cholesterol uptake/efflux and lipid trafficking expressed in mouse βtc3 cell line and human isolated islets of Langerhans. (**B**) RNA-seq data of genes involved in cholesterol homeostasis and expressed in mouse βtc3 cells (unpublished data). Data are expressed as reads per kilobase of transcript per million (RPKM) and are mean ± SD (n = 5 samples). (**C**) RNA-seq of genes involved in cholesterol homeostasis and expressed in human islets of Langerhans. Data are expressed as reads per kilobase of transcript per million (RPKM) and are mean ± SD (n = 6 healthy donors of varying BMI and age). The analysis on human islets was performed using the dataset published by Segerstolpe et al., 2016 [75].

**Figure 3 biomolecules-13-00224-f003:**
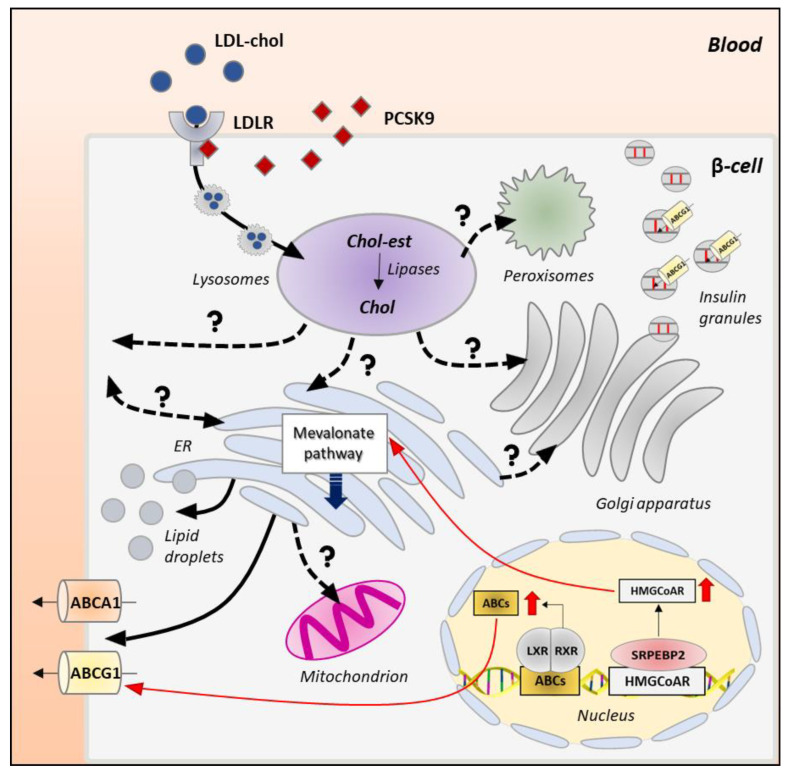
Intracellular itinerary of cholesterol in β-cells. Cholesterol is synthesized in the endoplasmic reticulum (ER) through the mevalonate pathway, which is controlled by the SREBP2-HMGCoAR axis (red arrow). Once synthesized, free cholesterol can be transported to the mitochondria, the Golgi apparatus and the plasma membrane; the mechanisms underlying cholesterol delivery to these organelles are completely unknown in pancreatic β-cells (dotted lines and question marks). Circulating LDL-cholesterol (LDL-Chol) is internalized in pancreatic β-cells by LDLR-mediated endocytosis and delivered to the lysosomes via clathrin-coated vesicles. A key regulator of this process is the proprotein convertase subtilisin/kexin type 9 (PCSK9), which routes LDLRs to the lysosomes for degradation, preventing their recycling to the plasma membrane. Once in the lysosomes, LDL-cholesterol (Chol-est) is metabolized by acid lipases to yield free cholesterol (Chol) that is transported to downstream organelles (plasma membrane, Golgi apparatus, peroxisomes and ER) through yet unknown mechanisms (dotted lines and question marks). Cholesterol accumulates also in insulin granules through ABCG1-mediated transport. Excessive free cholesterol is stored in lipid droplets or removed by ABCA1 and ABCG1 transporters, whose expression is controlled by LXR-RXR transcription factors (red arrow).

**Figure 4 biomolecules-13-00224-f004:**
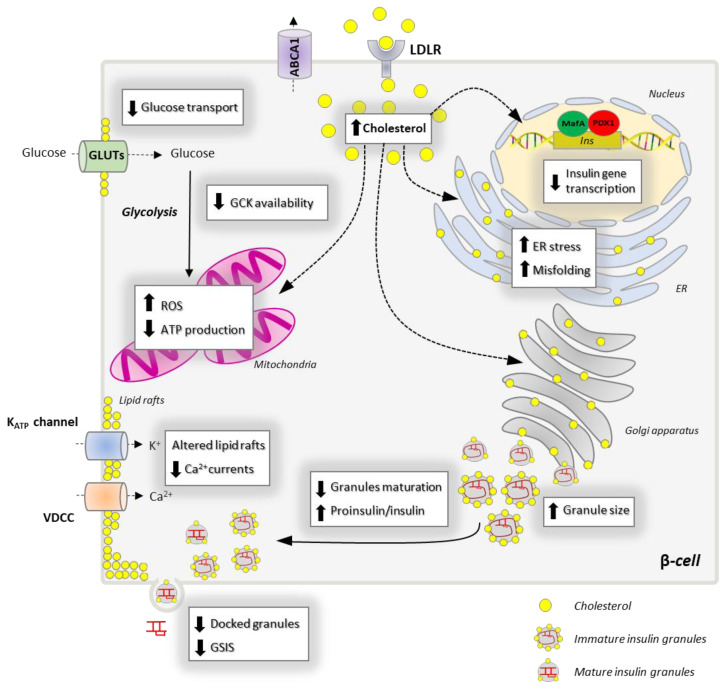
Impact of cholesterol overload on the insulin biosynthetic and secretive pathway. Increased cholesterol uptake via LDLR or reduced efflux through ABCA1 transporter leads to increased intracellular cholesterol, which strongly affects β-cell functionality through different mechanisms. Cholesterol accumulation downregulates the expression of PDX1 and MafA transcription factors reducing the transcription of the insulin gene. Cholesterol overload in the ER activates the UPR response leading to ER stress and proinsulin misfolding. Cholesterol also accumulates in the Golgi apparatus and in the membranes of insulin granules leading to their enlargement which, in turn, affects granules maturation (increased proinsulin/insulin ratio). Excessive cholesterol into the mitochondria induces the production of ROS, which impairs the mitochondrial membrane potential resulting in the reduction of ATP production. Cholesterol overload alters plasma membrane fluidity, lipid rafts composition and organization. These alterations result in reduced glucose transporters (GLUTs) activity and levels at the plasma membrane, decreased glucokinase (GCK) availability and altered distribution of voltage-dependent Ca^2+^ channels (VDCC) and K^+^_ATP_ channels. As a result, Ca^2+^ currents and granules fusion with the plasma membrane decrease, thus reducing the glucose stimulated insulin secretion (GSIS).

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
