# Peer review of "Cholesterol Redistribution in Pancreatic β-Cells: A Flexible Path to Regulate Insulin Secretion"

_biomolecules, 2023, doi:10.3390/biom13020224_

Round 1

Reviewer 1 Report

This review by Galli et al. described cholesterol homeostasis and trafficking and  the effects of cholesterol on insulin biosynthesis and secretion and provides an updated information of the state of the art on this issue. The authors describe and provide readers with interesting findings on the intracellular behavior of cholesterol and insulin. However, there are several concerns that need to be resolved.

1.  The cited references should match what is stated in the text.
       e.g. lines 585-589, Hussain and ref. no. 116,
       line 595, ref. no. 173, and so on.
 So, the entire text and all references must be reviewed.

2.  The information in the text needs to be reconfirmed.
       e.g. Lines 617—624, is ABCA1 correct?

3. In the text, Figure 2 and Figure 3 first appear on line 337 and line 332, respectively. Thus, Figures 2 and 3 should be exchanged.

4.  Figure 1
・If preproinsulin means mRNA, it should be drawn in such a way that proteins and mRNAs can be distinguished.

・Insulin and proinsulin have three disulfide bonds, but only two join the A and B chains.

・Somewhere in Figure 1, Insulin should also be indicated by a written description like folded proinsulin.

・Signaling pathways for k+, depolarization, and calcium influx should also be noted with arrows, etc.

・There are Box and BOX in figure 1 and Box in the legend, the description should be unified.

・In BOX1, the nuclear membrane should be removed from the enclosure.

・In Box2 and BOX2, the cell membrane and insuline granule are upside down and need to be oriented the same way.

5.  Figure 2
・A) and C)
Human gene symbols are all in upper-case and mice gene symbols are only the first letter in upper-case. This should be taken into account when writing.

・B)
The results in Segerstolpe et al., 2016 [77] are reported using human samples; are there results for βTC3 cells in Segerstolpe et al., 2016?

6.  Figure 3
In the figure legend, it should be noted in the description what Chol-est and chol represent.

Author Response

We would like to thank the reviewer for his/her in-depth comments, suggestions, and corrections, which were very helpful for improving the manuscript. Please find enclosed the point-to-point answers.

  1. The cited references should match what is stated in the text.
    e.g. lines 585-589, Hussain and ref. no. 116,
           line 595, ref. no. 173, and so on.
     So, the entire text and all references must be reviewed.

We would like to thank the reviewer for raising this point. We critically revised the references and changes have been highlighted in yellow in the text.

  1. The information in the text needs to be reconfirmed.
    e.g. Lines 617—624, is ABCA1 correct?

We would like to thank the reviewer for this observation. ABCA1 was correct in that lines, but the references were not. We modified them accordingly.

  1. In the text, Figure 2 and Figure 3 first appear on line 337 and line 332, respectively. Thus, Figures 2 and 3 should be exchanged.

Thank you for this observation. Figure 2 now appears on line 211 and Figure 3 on line 334.

4.  Figure 1
・If preproinsulin means mRNA, it should be drawn in such a way that proteins and mRNAs can be distinguished.

Thank you for this observation, pre-proinsulin mRNA and protein have been drawn in grey and dark red, respectively.

・Insulin and proinsulin have three disulfide bonds, but only two join the A and B chains.

Thank you for pointing this out. The suggested changes have been made.

・Somewhere in Figure 1, Insulin should also be indicated by a written description like folded proinsulin.

Following the reviewer’s suggestion, a legend describing the symbols used in the figure was added (box in the upper part).

・Signaling pathways for k+, depolarization, and calcium influx should also be noted with arrows, etc.

As suggested by the reviewer, we added arrows connecting K+ channel closure with β-cell depolarization, Ca2+ influx and insulin granules release.

・There are Box and BOX in figure 1 and Box in the legend, the description should be unified.

We would like to thank the reviewer for the observation. “Box” now appears only in lower case in the figure.

・In BOX1, the nuclear membrane should be removed from the enclosure.

As suggested by the reviewer, the nuclear membrane was excluded from the enclosure of Box1.

・In Box2 and BOX2, the cell membrane and insuline granule are upside down and need to be oriented the same way.

According to the reviewer’s suggestion, we changed Box2. Insulin granule and cell membrane are now oriented in the same way in the figure and in the box.

5.  Figure 2
・A) and C)
Human gene symbols are all in upper-case and mice gene symbols are only the first letter in upper-case. This should be taken into account when writing.

Thank you for this observation. Human gene symbols are now in upper-case both in the table and the histogram.

・B)
The results in Segerstolpe et al., 2016 [77] are reported using human samples; are there results for βTC3 cells in Segerstolpe et al., 2016?

We would like to thank the reviewer for raising this point. In Figure 2, results shown in panel B refer to βtc3 cells and were obtained by our laboratory (RNA-seq analysis), while those shown in panel C refer to human islets and derived from the analysis of Segerstolpe’s database. To avoid confusion, the following sentence was added to the figure caption: “The analysis on human islets was performed using the dataset published by Segerstolpe et al., 2016.“

  1. Figure 3
    In the figure legend, it should be noted in the description what Chol-est and chol represent.

Following the reviewer’s observation, the abbreviations (Chol-est and Chol) were defined in the figure caption.

Reviewer 2 Report

This is an exceptionally well-written review on an interesting topic regarding the role of cholesterol in regulating beta-cells. The discussion flows logically and is of interest not only for a specialized audience, but also for clinicians. The paper also benefits from very informative figures.

Author Response

We would like to thank the reviewer for her/his thorough manuscript review and enthusiastic comments. We are very pleased with the appreciation. 

Reviewer 3 Report

This is a very beautiful review article about how cholesterol homeostasis influences pancreatic β-cell function. Many metabolic disorders include inappropriate insulin secretion and unbalanced cholesterol levels. This review paper not only introduced insulin biosynthetic and secretive pathway, but also described the regulation of cholesterol in pancreatic β-cell. In addition, the author thoroughly explained the role of cholesterol in regulating insulin secretion. This innovative topic about how cholesterol mediates insulin secretion of β-cell has the potential to broaden our view as well as contribute understanding in research of diabetes and obesity.

The manuscript was written in very fluent English and contained detailed information. The only recommendation is to add a Figure in Section 5. “Effects of cholesterol imbalance on the insulin biosynthetic and secretive pathway”, thus readers can more easily grasp how cholesterol controls insulin secretion.

Author Response

We would like to thank the reviewer for his/her suggestion that has been included in the text. A new Figure 4 reporting the effects of cholesterol overload on β-cell functionality was inserted at the beginning.

Reviewer 4 Report

The review is in general well written and with good detail on cellular pathways and signaling concerning the importance of uptake, efflux and metabolism of cholesterol in beta cells, an interesting field not fully understood.

Some minor revisions require revision:

      1.  Line 38. There is evidence that the beta cell is not the only place in the body that produces insulin. It seems also to occur in the cerebral cortex (Csajbók, É.A., Tamás, G. Cerebral cortex: a target and source of insulin?. Diabetologia 59, 1609–1615 (2016). https://doi.org/10.1007/s00125-016-3996-2) ( GABAergic neurogliaform cells represent local sources of insulin in the cerebral cortex. J Neurosci. 2014 Jan 22;34(4):1133-7. doi: 10.1523/JNEUROSCI.4082-13.2014.). However, the pancreatic secretion is the only with a significative systemic action. For this reason, I advise modifying the assertion that insulin is only produced in the beta cell.

2. Table 2: It would be better for Table 2 to be located just after the text that quotes it, so that the reader can follow the order of the story in the review. It is quoted for the first time in line 377.

3.  The intracellular itinerary of cholesterol in beta cells and their regulation is well explained in the text. However, some details must be attended in figure 3.  The regulation of uptake and efflux must be better depicted since the effect of SRPEB2 and LXR/RXR in the nucleus is not completely shown. Also, the lipid droplet is shown but not named in the figure.

4. 4. In line 379, a potential mechanism of cholesterol transport between membranes, through MC is described and as example, the transportation of PS from ER to mitochondria through MAMs is named. But the authors do not clarify if there is a concrete example of this type of transport specifically on cholesterol in the beta cells.

Author Response

We would like to thank the reviewer for his/her thoughtful review of the manuscript. He raised important issues and inputs helpful for improving the manuscript. Please find enclosed the point-to-point answers.

Some minor revisions require revision:

  1. Line 38. There is evidence that the beta cell is not the only place in the body that produces insulin. It seems also to occur in the cerebral cortex (Csajbók, É.A., Tamás, G. Cerebral cortex: a target and source of insulin?. Diabetologia 59, 1609–1615 (2016). https://doi.org/10.1007/s00125-016-3996-2) (GABAergic neurogliaform cells represent local sources of insulin in the cerebral cortex. J Neurosci. 2014 Jan 22;34(4):1133-7. doi: 10.1523/JNEUROSCI.4082-13.2014.). However, the pancreatic secretion is the only with a significative systemic action. For this reason, I advise modifying the assertion that insulin is only produced in the beta cell.

We would like to thank the reviewer for this observation. We revised the text accordingly modifying the sentence as follows: “Despite reports indicating insulin expression in tissues such as the brain or liver (Csajbók and Tamás, 2016; Dong et al., 2001), the pancreatic β-cells remain the major source of insulin biosynthesis and secretion, therefore their dysfunction and/or death result in decompensated glucose homeostasis and diabetes development [3,4].”

  1. Table 2: It would be better for Table 2 to be located just after the text that quotes it, so that the reader can follow the order of the story in the review. It is quoted for the first time in line 377.

According to the reviewer’s suggestion, Figure 2 now appears after the text that quotes it.

  1. The intracellular itinerary of cholesterol in beta cells and their regulation is well explained in the text. However, some details must be attended in figure 3.  The regulation of uptake and efflux must be better depicted since the effect of SRPEB2 and LXR/RXR in the nucleus is not completely shown. Also, the lipid droplet is shown but not named in the figure.

We apologize for the missing details. As suggested by the reviewer, lipid droplets and details regarding the SREPB2/HMGCoAR and LXR-RXR/ABCs pathways (red arrows) were added in the figure.

  1. In line 379, a potential mechanism of cholesterol transport between membranes, through MC is described and as example, the transportation of PS from ER to mitochondria through MAMs is named. But the authors do not clarify if there is a concrete example of this type of transport specifically on cholesterol in the beta cells.

We would like to thank the reviewer for raising this point. To the best of our knowledge, the PS exchange between the ER and the mitochondria through MAMs has not been demonstrated yet in pancreatic β-cells. Therefore, to avoid confusion we revised the text as follow: “Even though this transport has not been identified yet in pancreatic β-cells, several lines of evidence recently suggested that MAMs occur also in these cells and are involved in the control of insulin secretion [117,118]. Whether they also play a role in the intracellular redistribution of cholesterol is not known but, interestingly, alteration of their organization has been detected in T2D and T1D scenarios [119–122]. Furthermore, palmitate treatment of Min6-B1 cells resulted in altered GSIS, elevated ER stress and decreased ER-mitochondrial contacts, potentially linking interorganellar interactions, insulin secretion and lipotoxicity [119]. Fatty acid-mediated ER damage and cholesterol dysregulation in β-cells are concisely highlighted later in this review (see section 5.1).“

Round 2

Reviewer 1 Report

The manuscript has been revised well. I think this manuscript will be   suitable for publication in Biomolecules.